# Short Commentary on Marine Productivity at Arctic Shelf Breaks: Upwelling, Advection and Vertical Mixing

Achim Randelhoff[1,*] and Arild Sundfjord[1]

[1]Norwegian Polar Institute, Fram Centre, N-9296 Tromsø, Norway
*now: Québec-Océan and Takuvik, Département de biologie, Université Laval, Québec, Canada
*Correspondence to:* Achim Randelhoff (achim.randelhoff@takuvik.ulaval.ca)

**Abstract.**

The future of Arctic marine ecosystems has received increasing attention in recent years as the extent of the sea ice cover is dwindling. Although the Pacific and Atlantic inflows both import huge quantities of nutrients and plankton, they feed into the Arctic Ocean in quite diverse regions. The strongly stratified Pacific sector has a historically heavy ice cover, a shallow shelf and dominant upwelling-favourable winds, while the Atlantic sector is weakly stratified, with a dynamic ice edge and a complex bathymetry. We argue that shelf break upwelling is likely not a universal but rather a regional, albeit recurring feature of "the new Arctic". Instead, it is the regional oceanography that decides its importance through a range of diverse factors such as stratification, bathymetry and wind forcing. Teasing apart their individual contributions in different regions can only be achieved by spatially resolved timeseries and dedicated modelling efforts. The Northern Barents Sea shelf is an example of a region where shelf break upwelling likely does not play a dominant role, in contrast to the shallower shelves north of Alaska, where ample evidence for its importance has already accumulated. Still, other factors can contribute to marked future increases in biological productivity along the Arctic shelf break. A warming inflow of nutrient-rich Atlantic Water feeds plankton at the same time as it melts the sea ice, permitting increased photosynthesis. Concurrent changes in sea ice cover and zooplankton communities advected with the boundary currents make for a complex mosaic of regulating factors that do not allow for Arctic-wide generalizations.

## Introduction

Surface waters throughout most of the world ocean are generally low in nutrients. In order to sustain primary production, new nutrients are required. These can come by means of mineral-rich rivers draining into coastal areas, turbulent small-scale mixing where underlying waters are rich in nutrients, upwelling of deeper nutrient rich waters, or even nitrogen fixation by some bacteria. In fact, upwelling in certain coastal areas and at shelf breaks in many regions of the world ocean supports intense marine production and can sustain rich regional fisheries (see e.g. Kämpf and Chapman, 2016). Where upwelling occurs, it is

often intimately linked to specific weather and climate patterns, such as storms (cyclones), or wind blowing from a preferential direction. The basic concept is that the winds set up spatially varying surface transport or forces surface water away from the coast, creating a divergence that draws up deeper waters which would otherwise be too heavy to be brought up by vertical mixing alone (Kämpf and Chapman, 2016).

5     Shelf break upwelling has recently received increasing attention also in the Arctic Ocean (Carmack and Chapman, 2003; Arrigo and van Dijken, 2015; Williams and Carmack, 2015, and more references below; for an overview of the geography, see Fig. 1). As the ice edge recedes from the shelves into the basin further and further each year (e.g. Stroeve et al., 2012), net primary production has been oberved to have increase Arctic-wide (Arrigo and van Dijken, 2011; Bélanger et al., 2013; Arrigo and van Dijken, 2015): Not only would less ice allow more solar radiation into the ocean, providing more of a scarce

10   requirement for photosynthesis. It is also assumed that winds can move the surface waters more effectively and lead to more pronounced shelf break upwelling (Carmack and Chapman, 2003), another flavour of the Arctic as that region of the world where the impacts of climate change are most pronounced.

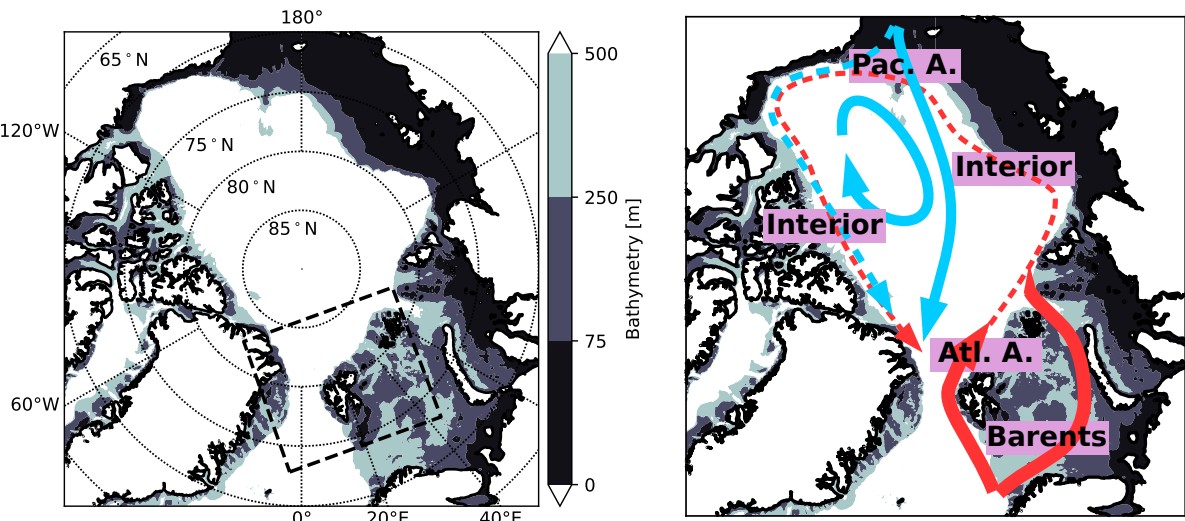

**Figure 1.** Map of the Arctic Ocean (based on Jakobsson et al., 2012), indicating the general geographic regimes. Left: Bathymetry of the shelf and shelf break area. The box in dashed lines shows the area in Fig. 4, left panel. Right: The Pacific Arctic, Atlantic Arctic, interior shelves (following Williams and Carmack, 2015); and the Barents Sea. Arrows show selected patterns of the general circulation (after Polyakov et al., 2012). Blue arrows: Pacific-derived and other freshwater flowing along the shelf break, through the Transpolar Drift and in the Beaufort Gyre. Red arrows: Atlantic-derived water entering the Arctic Ocean through Fram Strait and the Barents Sea, submerging north of the Barents Sea and recirculating along the shelf break through the Arctic Ocean. Other major currents are not indicated here as they are of minor importance to this paper.

## Upwelling in the Arctic

In their seminal 2003 paper mentioned above, Carmack and Chapman applied a numerical model to study shelf-basin exchange on the Beaufort Sea shelf and argued that decreased ice concentrations will enhance upwelling in the area. The argument goes like this: When a thick ice cover lies like a lid on the ocean, it absorbs most of the wind stress instead of transferring it to the underlying ocean. When the ice edge recedes far enough north that the shelf break is exposed, however, the winds can move around the surface waters more easily. Sustained easterlies, for example, will lead to a northward Ekman transport, and where the shelf is shallow enough that it affects surface currents (see Fig. 2 and 3), deeper waters are drawn up to balance the off-shelf transport.

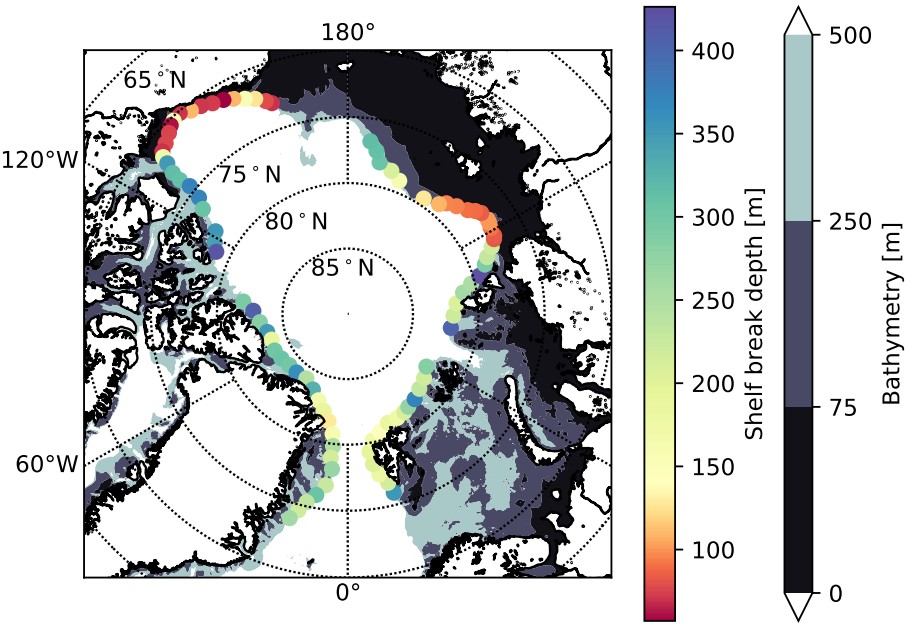

**Figure 2.** Depth of the Arctic shelfbreak extracted from the IBCAO V3 bathymetry of the Arctic Ocean (Jakobsson et al., 2012), excluding only the Saint Anna Trough and the Chukchi Borderland. Most visible are the continental shelf off Alaska and the westernmost part of the Canadian shelf, where Carmack and Chapman (2003) conducted their study and upwelling has been frequently documented, and north of the Laptev Sea. In most other areas of the Arctic Ocean, the shelfbreak is several hundred meters deep and therefore out of reach to interact significantly with Ekman-driven surface ocean dynamics. For a detailed explanation of the algorithm and the computer code used to extract shelfbreak depths, see the supplementary material.

This argument was reinforced by a number of studies conducted in the Pacific Arctic (Williams et al., 2006; Schulze and Pickart, 2012; Spall et al., 2014; Arrigo et al., 2014; Lin et al., 2016), which directly extended earlier direct observations of shelf break upwelling dating back to at least the 1980s (e.g. Aagaard et al., 1981). A detailed study (Spall et al., 2014) on the

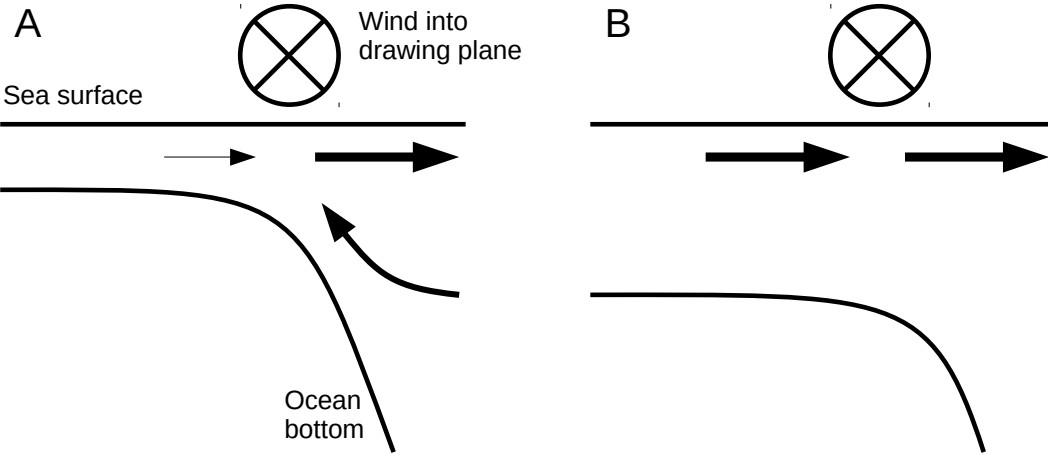

**Figure 3.** Schematic of the mechanism behind wind-driven shelf break upwelling. When wind blows along the shelf break, it generates an Ekman current (horizontal arrows) off-shelf. A: When the shelf is shallow enough, the current over the shelf is slowed down, leading to a horizontal divergence and thus pressure gradient that is filled by drawing up deeper waters. B: When the shelf is deeper, there is no horizontal divergence. Other mechanisms, such as dynamic uplift, are independent from wind and not discussed here, but see e.g. the book by Kämpf and Chapman (2016).

dynamic response during one particularly impressive example of shelf break upwelling in the Chukchi Sea (Arrigo et al., 2014) demonstrated potentially large contributions to primary productivity in that area.

The idea has since caught on to explain or project marine productivity also in other regions of the Arctic Ocean, for example at the Barents Sea shelf break. There it has appeared both in numerous personal communications among the community
5  working with the physical and ecological environment of the Barents Sea, as well as a number of published articles (see e.g. Falk-Petersen et al., 2014; Tetzlaff et al., 2014; Wassmann et al., 2015; Hunt et al., 2016; Våge et al., 2016; Haug et al., 2017). Thus it might appear as if shelf break upwelling is currently being cemented as a universal paradigm to conceptualize *the* "new" Arctic Ocean where global climate change is taking us. We will argue that some of the regional differences cannot be ignored when discussing what governs productivity in the various shelf regions.

10 **Many interconnected phenomena**

Upwelling comes in many different forms: The well-known upwelling that feeds so many productive coastal areas of the world is created by winds blowing along-shore, driving an offshore surface current that "pulls up" nutrient rich waters. (This will in practice most often be the Ekman transport; however, shelf break upwelling would function in much the same way at the equator where there is no Coriolis force, even though upwelling-favourable winds would then blow directly off-shelf instead of

along-shelf.) The divergence sets up a horizontal gradient in sea surface height that balances the Coriolis force, meaning that deeper waters are drawn towards the surface and/or onto the shelf (again, see e.g. Kämpf and Chapman, 2016).

Alternatively, storms can lift deeper waters up to the shelf break, making them spill over and mix with shelf waters. Canyons and troughs that cut into a continental shelf may aid by steering the flow there through its topography. All of these phenomena can act together to bring new nutrients into shelf waters.

But besides upwelling, other factors are at play. Two important ones are vertical mixing and advection with large scale ocean currents, and both of them can become entangled with upwelling in that they can lead to similar effects in the regional oceanography and be hard to tell apart by the most basic means of hydrography which are vertical profiles of temperature and salinity. Because different areas within the Arctic Ocean are subject to very different forcing, large gradients in physical properties exist between e.g. Bering Strait, Fram Strait and the Siberian Shelf. Naturally, this means that also the drivers of marine productivity will vary strongly between these areas.

### Drivers of marine productivity vary across the Arctic Ocean

There is an ample storage of freshwater in the Arctic Ocean because of the large rivers draining Siberia and North America, but also because the inflow of Pacific Water through Bering Strait is much fresher than its Atlantic counterpart (Aagaard and Carmack, 1989). But the freshwater is not evenly distributed: Most of it is found in the Beaufort Gyre located around the Canadian Basin (e.g. Morison et al., 2012; Proshutinsky et al., 2015). When light water (at low temperatures, this means fresher) sits on top of heavy water, mixing will not be as efficient (e.g. Osborn, 1980), which means that the most important factor for vertical mixing is vertical stability (since overall, there is a given amount of energy available to stir the ocean, e.g. from tides, wind and so on.) In the Beaufort Gyre, all the freshwater and the resulting strong stratification severely restrict the upward supply of fresh nutrients, making it one of the most nutrient-depleted regions of the world ocean (Gruber and Sarmiento, 1997; Codispoti et al., 2013; Tremblay et al., 2015).

In contrast, the Atlantic inflow along the shelf break north of Svalbard is much denser than the surface waters of the central Arctic Ocean, but nevertheless extends up to the surface (see e.g. Rudels, 2016, ; an illustration is also given in Fig. 4). Seeing this situation in the contour plot of a hydrographic transect (see right panel of Fig. 4) may at first look like a classical upwelling scenario: Surely there must have been upwelling to get the heavy waters up there in the first place? The answer is that not necessarily - what we are seeing is Arctic and Atlantic water masses meeting, and the narrow but strong gradient is maintained by a continuous inflow of more Atlantic Water. In the absence of detailed (hydrographic) timeseries, it is impossible to say anything conclusive about the state of upwelling from the right panel of Fig. 4 alone.

We thus need to distinguish between basin-scale and regional hydrography, that is between strong haline stratification in the Arctic Ocean in general and weak thermal stratification in the Atlantic inflow (see the distinction between "alpha" and "beta" oceans as in Carmack, 2007). The salient point is this: As the Atlantic Water is cooled on its way north, it loses stability, potentially leading to wintertime convection (Ivanov et al., 2016) and efficient vertical mixing. The result is that the surface layer nutrient reservoirs are replenished long before the end of winter (Randelhoff et al., 2015); increased wintertime upwelling

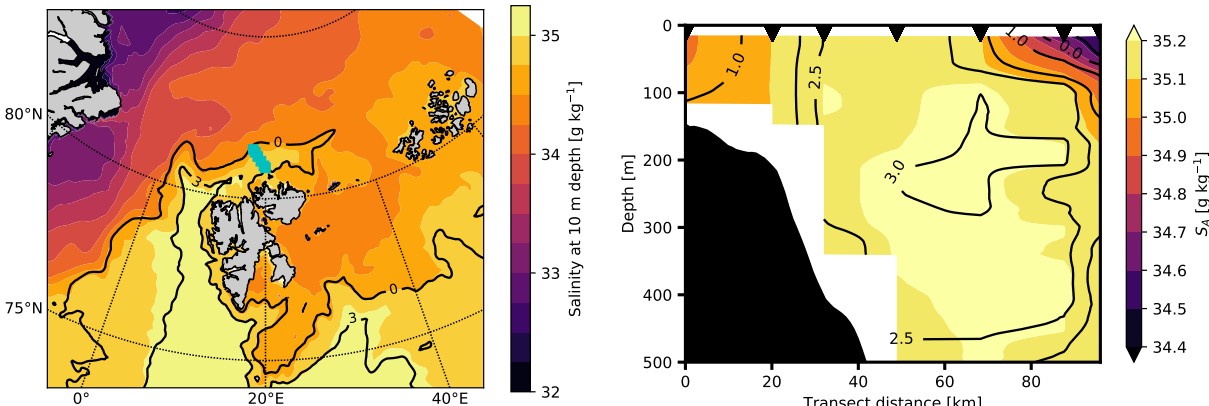

**Figure 4.** Representative illustration of the hydrographic regime in the Atlantic inflow area along the northern Barents Sea shelf break. Left: Inflowing warm and salty Atlantic Water maintains high surface salinity on and around the shelf, enabling convection when the surface waters are cooled in winter. Color scale shows salinity, black contour lines show the 0 and 3°C isotherms. (Average of 2015-2018 monthly January mean at 10 m depth from the operational ocean reanalysis Mercator, downloaded from http://marine.copernicus.eu. The version of Mercator used for this plot is a global ocean forecasting model on a $1/12° \times 1/12°$ grid and showed good agreement with winter data for this area in a study by Koenig et al. (2017).) The cyan line indicates location of transect displayed in right panel. Right: Seawater absolute salinity $S_A$ and conservative temperature in a typical wintertime transect across the shelf slope north of Svalbard, sampled in January 2014 (data from approximately 81.5°N, 17.5°E, RV Helmer Hanssen, Carbon Bridge project (Randelhoff and Sundfjord, 2017, published dataset; see also Randelhoff et al., article submitted to Frontiers in Marine Science); see left panel for location). Salinity is plotted on the color scale, temperature is marked (in °C) on the black isolines inside the plot. The surface water is markedly heavier above the upper shelf slope than over the deep basin. Black triangles mark hydrographic stations. The black patch marks the along-transect bathymetry extracted from the IBCAO V3 bathymetry (Jakobsson et al., 2012). During the sampling of this transect, winds were moderate southerlies to south-south-easterliers, so mean Ekman transport in the surface was mainly directed along-shelf to the east.

will not bring more nutrients to the surface. Essentially, the upwelling water mass would have the same salinity and nutrient characteristics as the one that is already present in the surface; upwelling does not add nutrients when there is no vertical gradient in nutrient concentrations. In contrast, the Beaufort sea is strongly stratified throughout the year; if winter upwelling is to increase there because of reduced sea ice, this can be an important factor contributing to the pre-bloom nutrient pool.

5      In contrast to storms, which can lift deeper waters independently from any sort of topographic constraint (i.e. Ekman pumping), coastal and shelf break upwelling driven by specific wind directions need the presence of a coastline or a sufficiently shallow shelf. This is because it requires a horizontal divergence in the off-shelf transport of surface waters. This divergence can only be potent enough when the shelf itself is shallow enough to actually constrict the surface flow over the shelf (Fig. 3). Whereas large swaths of the continental shelves of the Arctic Ocean are very shallow (in parts less than 50 m), the Northern

10    Barents Sea shelf break is relatively deep at around 150-250 m (see Fig. 2). Because surface and bottom boundary layers will not overlap in this case (common values for Ekman layer depth in the literature are few tens of meters, see Price and Sunder-

meyer, 1999), shelf break upwelling as an effect of along-shore winds is presumably negligible. (Also note that Ekman layer depth decreases with increasing latitude and decreasing wind strength (Wang and Huang, 2004), and that during the stratified summer period, the Ekman layer will at any rate be restricted to at most the surface mixed layer, see e.g. Price et al. (1987)).

In general, the regions that (only based on the depth of the shelf break) stand out as most prone to wind-driven shelf break
upwelling are the aforementioned continental shelves of Alaska and the westernmost part of northern Canada, and possibly the Laptev Sea, although the shelf is rather wide here, potentially diminishing the effect of easterlies somewhat. In regions where the shelf is narrow, the presence of the coastline can aid in upwelling of deeper waters. Seeing that the Chukchi and Siberian shelves are rather wide, potential upwelling will likely be relatively weak across large swaths of the Arctic shelf regions.

**Summertime upwelling north of Svalbard?**

We have seen how the pre-bloom surface nutrient inventory at the northern Barents Sea shelf break can be replenished just by the inflowing Atlantic water, without recurrence to wintertime upwelling. In summer, however, nutrients are depleted in surface waters, such that even sporadic upwelling could inject nutrients that could be utilized immediately and funneled into the food web (see e.g. Ch. 3.2, Kämpf and Chapman, 2016).

Here, another difference between the Atlantic and Pacific inflow areas comes into play, namely dominant wind patterns: The
Beaufort Sea shelf is dominated by the Beaufort High–Aleutian Low system, meaning predominantly easterlies at the Canadian shelf break (e.g. Serreze and Barrett, 2011). The atmospheric circulation in the Atlantic sector is more dynamic in summer, with less of a preference for a specific upwelling-favourable wind direction (see e.g. Fig. 5). This comes on top of a general pattern where wind speeds north of Svalbard are lower in summer than in winter. Fig. 5 illustrates how only roughly 2% of all summer days through the last 30 years can be considered upwelling-favourable, using a very generous criterion for what
constitutes "upwelling-favourable", and even this is assuming that the local topography would allow for this kind of upwelling. (Again, note the difference to the Beaufort shelf, where winds are very much upwelling-favourable also in June, see Lin et al. (2016).) There might still be storms that make deeper waters spill onto the shelf by Ekman pumping alone, but also these have a tendency to occur more frequently in the winter season (see also Lind and Ingvaldsen, 2012).

As has been shown above, wind statistics as well as general physical considerations and geographical features - the northern
Barents Sea shelf being too deep for surface and bottom Ekman layers to overlap and produce shelf break upwelling - imply that upwelling should not be expected to feature very prominently on the Barents side of the Arctic. This is not to say that upwelling events cannot ever happen (and indeed, in a system as complex as the Earth, it would be surprising if it would never happen), but no known physical mechanism would suggest a magnitude, frequency or importance similar to what has been found in the Pacific sector. To illustrate our point, we refer to recent analysis by A. Renner and collaborators. They have
analysed the first year-long time series from a moored CTD array over the shelf slope north of the Barents Sea (A-TWAIN project, at 30°E). Applying methods that have successfully detected frequent occurrence of upwelling over the Beaufort Sea slope (Lin et al., 2016), they could not identify signatures of upwelling in the density field in response to possibly favourable along-slope winds (A. H. H. Renner, pers. comm. and article in review for Journal of Geophysical Research).

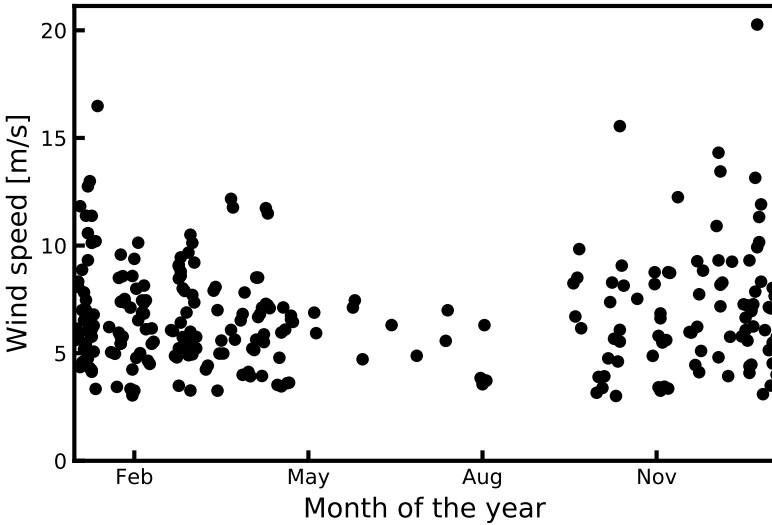

**Figure 5.** Days of "potentially upwelling-favourable" winds north of Svalbard 1987-2017 assuming the local bathymetry facilitates such upwelling, based on ERA-INTERIM data (Dee et al., 2011) for the region 79–81°N, 5–30°E. A daily windspeed was considered "potentially upwelling-favourable" if its (approximately easterly) along-shelf component exceeded 3 m s$^{-1}$ for at least 3 consecutive days. (3 m s$^{-1}$ is rather low a wind speed, well below the "optimal environmental window" of 5–6 m s$^{-1}$ for upwelling suggested by (Cury and Roy, 1989). and makes for a generous criterion in this regard. Likewise, Kämpf and Chapman (2016, Ch. 2.1) give a timescale of around 5 days from the onset until the complete development of coastal upwelling. Effectively, both criteria should err on the upwelling-favourable side.) From the beginning of May through August each year, ∼2% of all days were "potentially upwelling-favourable".

**Climate Change and the Future of Arctic Marine Productivity**

Shelf break upwelling is often thought to become more prominent in the Arctic as the ice recedes poleward with ongoing climate change, exposing the shelf break more and more (see references given in the previous section "Upwelling in the Arctic"). But it should be kept in mind that the mere earlier presence of an ice cover would not have prohibited wind driven upwelling (or Ekman pumping for that sake), and could even have enhanced it in some circumstances. For instance, Martin et al. (2014) showed how a loose ice cover (80–90% ice concentration) can yield an optimum transfer of wind energy into the upper ocean when internal ice stresses are negligible, seeing that sea ice has a rougher surface than open water and can therefore be moved around more easily by the winds. This is consistent with the observation of Schulze and Pickart (2012) that the upwelling response at the Beaufort Sea shelf off Alaska was strongest when there was partial ice cover. Once again, there are differences between the historically thick, multiyear ice cover of the Pacific Arctic (Maslanik et al., 2007) and the more dynamic first- and second year ice cover north of Svalbard (Renner et al., 2013). In the latter area, it is not a new feature that the ice cover is quite dynamic and rough, which possibly leads to an efficient transfer of wind energy as was demonstrated in the previously mentioned paper by Martin et al. (2014). It is therefore not a given that reduced ice cover north of Svalbard

automatically will make surface currents more responsive than they were in the past, especially under the responsive summer pack ice, when upwelling would have the chance to substantially alter the marine ecosystem through sporadic nutrient input.

In fact, there are pathways entirely unrelated to upwelling through which climate change probably is impacting and enhancing marine productivity. Indeed, the regional loss of sea ice has been attributed to inflow of warmer Atlantic Water (Onarheim et al., 2014). As the Atlantic Water travels further and further east along the shelf break before it is sufficiently cooled and its core is subsequently subducted under the Arctic water masses, it pushes back the ice edge and erodes stratification (Polyakov et al., 2017) – meaning it provides access to nutrients and light at the same time! This will enhance regionally averaged primary production by itself, without the need to invoke shelf break upwelling.

In addition to heat, salt and nutrients, the Atlantic (like the Pacific) water also carries large amounts of zooplankton. This makes the inflow areas perfect feeding grounds for larger fish and mammals, adding onto local primary production. For instance, there is an excess of organic carbon production NW of Spitsbergen in May and June (Maria Vernet, pers. comm.), in agreement with modelling results (e.g. Wassmann et al., 2015). As sea ice recedes north- and eastward, it might extend this region of net heterotrophy (carbon consumption). However, results from a coupled ocean and ecosystem model indicate that by the end of the 21st century, zooplankton advection along the shelf break will dwindle, and marine life in the area might rely much more on local production (Wassmann et al., 2015). Such processes would contrast a projected pan-Arctic strengthening of upper ocean stratification that might lead to a smaller plankton size-spectrum, fuelling a food web that recycles more than providing food for higher trophic levels (e.g. Li et al., 2009, 2013).

**Summary and Conclusions**

Detailed measurements and analyses with spatial and temporal resolution are necessary in order to detect upwelling in general; shelf break upwelling in the Arctic is no exception. In general, moored CTD arrays in conjunction with wind data are a solid foundation to detect upwelling in the field; hydrographic snapshots are rarely enough to establish its dynamics and drivers. The 2-dimensional modelling approach of Spall et al. (2014) has proven particularly valuable for mapping out upwelling-driven nutrient transport across the Beaufort Sea shelf break, and a similar model could yield essential insight in other areas of the Arctic Ocean as well. Furthermore, the role of "dynamic uplift" (Kämpf and Chapman, 2016, Ch. 2.1) - where e.g. eddy shedding of a boundary current can lead to changes in its position onto the shelf - for shelf-basin exchange is not yet well understood in this area.

More generally, it would appear that changes in cross-shelf exchange are most important for the interior shelves (sensu Williams and Carmack, 2015) where nutrients are rather scarce to begin with. There is the projection that continued warming will release organic nutrients bound in the permafrost landscapes of northern Siberia and Alaska and flush them out into the Arctic Ocean (Frey and McClelland, 2009). Beyond these, rivers do not carry significant amounts of nitrate, one of the scarcest and most important mineral nutrients in the Arctic Ocean. Profound changes in the on-shelf transport of nutrient-rich water from the Atlantic Water boundary current might thus have big impacts on integrated productivity. Changes in the position of

the ice edge can also effect changing storm tracks and hence Ekman pumping. This too is a complex issue and there are no clear answers regarding its effect on nutrient transport onto the shelf.

Whatever the final result, Arctic marine life will find itself in a vastly different habitat within a tangible number of decades, showcasing the Arctic as a region where drastic changes are happening fast and, equally important, non-linearly. This also means that even dynamically isolated phenomena have to be evaluated against their specific regional backgrounds.

*Acknowledgements.* We thank Randi Ingvaldsen for very useful feedback and discussions on an earlier draft of the manuscript. AR was funded by the Norwegian Research Council project Carbon Bridge, a Polar Programme (project 226415) funded by the Norwegian Research Council.

*Competing interests.* The authors declare that no competing interests are present.

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
