# Peer review of "Short Commentary on Marine Productivity at Arctic Shelf Breaks: Upwelling, Advection and Vertical Mixing"

_Ocean Science, 2017_

## Referee Comment (RC1) · Anonymous Referee #1 · 13 Oct 2017

The authors made good points on explaining the shelf-break upwelling is regional phenomena and there are other factors need to be considered, such as the wind stress, stratification, vertical mixing and advection. The authors proactively cautioned crowning the shelf-break upwelling as a universal paradigm over new ice depleting Arctic Ocean.

1. Are the authors arguing a false impression? The self break upwelling itself is a regional phenomena. Did any of the papers that the authors refereed, Carmack and Chapman, 2003; Falk-Petersen et al., 2014; Wassmann et al., 2015; Våge et al., 2016; Haug et al., 2017), claimed that the shelf break upwelling is a "universal" phenomena?

[Figure]

2. The example that the authors gave, the Barents Sea, does not have favourable wind for the wind driven upwelling. Why the authors suggest that people will mistakenly think that the salinity front observed is a shelf break upwelling?

—————————————————————

---

## Referee Comment (RC2) · Anonymous Referee #2 · 8 Nov 2017

Review of "Short Commentary on Marine Productivity at Arctic Shelf Breaks: Upwelling, Advection and Vertical Mixing" by Achim Randelhoff and Arild Sundfjord

The authors discuss shelf break upwelling in the Arctic Ocean and argue that it should have different characteristics in different parts of the Arctic Ocean and not necessarily will be pronounced phenomena in the Atlantic sector of the Arctic Ocean in general and on the Barents shelf break in particular as a result of the climate change.

**Major comments**

It seems that authors try to argue with the opinion expressed in some of the recently published papers, that in the future conditions for the shelf break upwelling in the Atlantic sector of the Arctic will become more favourable. However, from the introduction (or rather "Upwelling in the Arctic" section) it is not clear what are the arguments authors fighting against. They mention personal communications and reference several papers, but did not provide any details.

The counter argumentation is very week. It is basically a collection of statements that are not supported by any evidence. It is just author's speculations on the topic with ideas that may or may not be true.

I have a hard time to define the type of this article and the purpose it is written for. The topic of the shelf break upwelling in Atlantic sector is very interesting and it would make a great contribution to our understanding of the Arctic Ocean when investigated properly though numerical modelling or data analysis. Unfortunately, this manuscript lacks any scientific novelty supported by evidence. I also believe it is too shallow to be a review. I do not recommend this manuscript for publishing in "Ocean Science".

**Minor comments**

P 2, L 2 It would be nice to provide references, showing that it "received increased attention". Now after the sentence you make a reference to the figure, which seem strange and out of place.

Fig. 1. Why you illustrate Atlantic Water inflow by snapshot from the model, that can be pretty far from reality (Hattermann et al., 2016 do not use data assimilation)? Why not from climatology or some reanalysis product (e.g MERCATOR OCEAN)?

P3, L5-7 You should really provide more evidence that this is now a "universal paradigm" and that the paper you mention above are actually directly and unconditionally transfer results obtained for the Pacific sector to the Atlantic Sector.

P3, L21 Gradients of what?

Fig. 2 Why you use this transect? Is it typical? Why not climatology or reanalysis?
P4, L4-9 It is not clear to me why any reader familiar with Arctic Ocean hydrography must think that Fig. 2 show typical upwelling situation? P4, L13 Ivanov et al., 2016 show that under certain conditions heat from the Atlantic Water can mix up to the surface, but this process is not constant and over the northern Barents Sea shelf the thermal stratification in the upper 100 meters is actually still quite strong most of the time.

P5 L3-15 Statements in this section need supporting evidence. Now it is pure speculations.

P5, L 17 No, we haven't. You just claim it to be true earlier, but you did not show anything to support this claim.

I am sorry but most of the rest of the analysis is again just pure speculations and to my opinion have no value as a review.

OSD

---

## Referee Comment (RC3) · Anonymous Referee #3 · 24 Nov 2017

Review Comments for "Short Commentary on Marine Productivity at Arctic Shelf Breaks: Upwelling, Advection and Vertical Mixing"

This short commentary manuscript brings up a hot and important topic on how wind-driven upwelling, in conjunction with dramatic sea ice loss, may affect ocean productivity on the vast Arctic continental shelves. The authors argued that "shelf break upwelling is likely not a universal but rather a regional, albeit recurring feature of the new Arctic." I do agree with the authors that regional geographic, atmospheric, oceanographic, and sea ice conditions must be taken into account when assessing pan-arctic upwelling phenomena and their impacts on upper ocean nutrient supply and primary

production processes. Nonetheless, the authors can better justify their arguments and greatly improve the paper by providing more concrete data analysis and evidence, particularly in the northern Barents Sea shelf where the authors claim upwelling may function differently from other Arctic shelf systems.

Other comments:

Figure 1: The illustrations are too vague and lack important geographic and hydrographic features. In the left panel, I would suggest the following changes: (1) add latitudinal circles and longitudinal lines, (2) use a better color map to illustrate bathymetry (or at least supplement a color bar for the grayscale), (3) draw general surface and bottom circulation patterns.

In section "Many interconnected phenomena": The authors tried to explain different physical mechanisms that drive upwelling and other physical processes that interact with upwelling. I am uncomfortable that the authors did not attempt to put their discussion in the context of rich literature in upwelling. There is not a single citation in the section, which is unusual.

In section "Drivers of marine productivity vary across the Arctic Ocean": The authors should provide observational evidence when claiming Beaufort Gyre region is "one of most nutrient-depleted regions of the world ocean".

Figure 2: Please mark corresponding transect in Fig 1 left panel. Why not plotting temperature, salinity and density fields in this transect all together so that readers can better interpret Atlantic and Arctic water masses, vertical mixing, thermal or haline stratification? How many CTD profiles were casted along this transect? Please mark the CTD cast locations. What were the wind conditions during this transect sampling? I think wind diagnosis would be critical in answering whether or not vertical mixing was caused by upwelling.

Figure 3: This is not an effective way to illustrate wind patterns. Did black dots represent speeds of east wind component? So only those with 3 m/s or more were showed in the figure? How often were the wind measured? The author stated "only 2% of all summer days through the last 30 years can be considered upwelling-favorable", but what's the sample size in total? It didn't look to me that 30-year wind measurements were included in the analysis given this few data points. To demonstrate seasonal differences in wind patterns and highlight the summer season, a plot of four wind roses that aggregate seasonal wind measurements might be more informative and illustrative.

In section "Summertime upwelling north of Svalbard?", the argument is unconvincing without showing results from mooring or ship-based hydrographic measurements. Personal communication is not sufficient.

In section "Climate Change and the Future of Arctic Marine Productivity", I think another relevant point is the changing phytoplankton abundance and species composition in response to changing hydrography and nutrients. I would suggest the authors to briefly touch on this point. Two examples are: 1) Li, W. K. W., F. A. McLaughlin, C. Lovejoy, and E. C. Carmack (2009), Smallest algae thrive as the Arctic Ocean freshens, Science, 326, 539; 2) Li, W. K. W., E. C. Carmack, F. A. McLaughlin, R. J. Nelson, and W. J. Williams (2013), Space-for-time substitution in predicting the state of picoplankton and nanoplankton in a changing Arctic Ocean, J. Geophys. Res. Ocean., 118(10), 5750–5759.

---

## Author Comment (AC1) · 19 Dec 2017

[Below, please find: Your comments preceded by RC, "Reviewer Comment", and our replies preceded by AR, "Author Reply"]

RC: The authors made good points on explaining the shelf-break upwelling is regional phenomena and there are other factors need to be considered, such as the wind stress, stratification, vertical mixing and advection. The authors proactively cautioned crowning the shelf-break upwelling as a universal paradigm over new ice depleting Arctic Ocean.

[Figure]

1. Are the authors arguing a false impression? The self break upwelling itself is a regional phenomena. Did any of the papers that the authors refereed, Carmack and Chapman, 2003; Falk-Petersen et al., 2014; Wassmann et al., 2015; Våge et al., 2016; Haug et al., 2017), claimed that the shelf break upwelling is a "universal" phenomena?

AR: > Thank you very much for reviewing our paper. Below, find our replies to your general comments.

> To our knowledge none of these papers claimed that it is a "universal" phenomenon, and neither did we claim that they had done so. However, based on the available literature, there are multiple claims that (and investigations whether) shelf break upwelling all across the Arctic enhances productivity. Calling this a "paradigm", a way of thinking, intends to say: Whenever someone talks about "Arctic shelf break" and "productivity", many people appear to associate that with "upwelling" and next "increasing upwelling leading to increased productivity due to receding sea ice".

RC: 2. The example that the authors gave, the Barents Sea, does not have favourable wind for the wind driven upwelling. Why the authors suggest that people will mistakenly think that the salinity front observed is a shelf break upwelling?

AR: > There are two issues here. To take your second point first, we are as puzzled as you are that this salinity front as such is sometimes considered to be due to shelf break upwelling. But it has happened and continues to do so; please see references given in the text (section "Upwelling in the Arctic").

> We are happy that you agree with us on the general point that not every front at an Arctic shelf break is indicative of upwelling. This is a pretty basic point, and taken at face value, we assume few would defend it in all its generality. However, as complex as the underlying physics are, we have felt the need to clarify its most salient aspects as a basis for discussions of (the even more involved issue of) biological productivity. We believe our paper is helpful for two audiences: People interested mostly in the biology, perhaps with less familiarity with the underlying physics, and people who do know the

physics but are looking into how this links to the biology.

> To your first point and eventually linking back to the point we were just making: The Barents Sea actually does (during wintertime, see our Figure 3) have winds that could, just based on alignment of the wind with the shelfbreak and coastline, drive upwelling if a number of additional factors are satisfied. What is dubious, however, is the link to biology, because, as we show based on available literature, nutrients are abundant in that region without any need for this wintertime upwelling. It is exactly the link to biology which is the focus of our paper, as we are not aware of any equally accessible, concise paper that lays out these kinds of processes.

---

## Author Comment (AC2) · 19 Dec 2017

[Below, please find: Your comments preceded by RC, "Reviewer Comment", and our replies preceded by AR, "Author Reply" and MC, "Manuscript Change"]

RC: Review of "Short Commentary on Marine Productivity at Arctic Shelf Breaks: Upwelling, Advection and Vertical Mixing" by Achim Randelhoff and Arild Sundfjord

The authors discuss shelf break upwelling in the Arctic Ocean and argue that it should have different characteristics in different parts of the Arctic Ocean and not necessarily will be pronounced phenomena in the Atlantic sector of the Arctic Ocean in general

and on the Barents shelf break in particular as a result of the climate change.

Major comments It seems that authors try to argue with the opinion expressed in some of the recently published papers, that in the future conditions for the shelf break upwelling in the Atlantic sector of the Arctic will become more favourable. However, from the introduction (or rather "Upwelling in the Arctic" section) it is not clear what are the arguments au- thors fighting against. They mention personal communications and reference several papers, but did not provide any details.

The counter argumentation is very week. It is basically a collection of statements that are not supported by any evidence. It is just author's speculations on the topic with ideas that may or may not be true.

I have a hard time to define the type of this article and the purpose it is written for. The topic of the shelf break upwelling in Atlantic sector is very interesting and it would make a great contribution to our understanding of the Arctic Ocean when investigated properly though numerical modelling or data analysis. Unfortunately, this manuscript lacks any scientific novelty supported by evidence. I also believe it is too shallow to be a review. I do not recommend this manuscript for publishing in "Ocean Science".

AR: > Thank you for taking the time to review our manuscript. Below, find our replies that we hope might help convince you regarding the purpose and quality of our "short commentary".

> First, about the type of article and intended readership. You correctly state that this paper lacks the novelty to be an original research article. It is also not a comprehensive review of every study that has been done on this issue (the choice "review article" in OSD is for practical reasons mostly, and suggested by the Editor). Rather, as the title states, it is a "Short Commentary" - implying we take a stance in a debate. (If you do not believe there is such a debate, please go to a relevant conference, mention Arctic shelf break upwelling, sit back and enjoy the discussion.)

> You mention readers "familiar with Arctic Ocean hydrography" would not hold some of the misconceptions we argue against. We can agree here, maybe pending some clarifications about what constitutes "familiar". But not everyone interested in the biological productivity in Arctic shelf regions is equally well-versed in the underlying physics. This readership in particular may profit from having all the contributing factors laid out in a single, accessible, quick-to-read piece of text, instead of wading through a lot of more detailed literature to acquire the familiarity with the material. (Something similar goes for physicists wishing to link their research to biology.) With the present manuscript, we tried to provide such an easily accessible text, which has not been published before as far as we know. Following our initial inquiry, we have been encouraged by the handling editor to submit such a text.

> Second, you repeatedly claim that the entirety of the manuscript is based on speculations and statements for which there is no evidence. You must certainly mean "support in the literature" or some such. To this effect, we have in the revised version added a good number of references that will hopefully clarify what we base our reasoning on, even if we initially (and mistakenly) thought these issues were too basic to have to reference them.

> Overall, you criticise us for not writing a different kind of article, which is fair enough, but misses the point of whether this article is useful to large enough an audience.

RC: Minor comments

AR: > [Below, please find: Your comments preceded by RC, our replies preceded by AR, and our edits to the text preceded by MC.]

RC: P 2, L 2 It would be nice to provide references, showing that it "received increased attention". Now after the sentence you make a reference to the figure, which seem strange and out of place.

AR: > The reference (after the words "Arctic Ocean") should have been to the left

panel of Fig. 1 only, which shows a map of the Arctic Ocean. We apologize and appreciate that you brought this to our attention so it could be clarified as it should. MC: > The reference now reads "see below for a list of references; for an overview of the geography, see the left panel of Fig. 1", and a later reference to Fig. 1 now reads "an illustration is also given in the right panel of Fig. 1".

RC: Fig. 1. Why you illustrate Atlantic Water inflow by snapshot from the model, that can be pretty far from reality (Hattermann et al., 2016 do not use data assimilation)? Why not from climatology or some reanalysis product (e.g MERCATOR OCEAN)?

AR: > As you say yourself, it is an illustration, and it does in fact capture the real-world features and patterns that are relevant for this paper (inflow of near-surface warm and salty water). We could have hand-drawn something (this, too, "pretty far from reality", but which would still capture the same essence), but for ease of producing the figure and because it looks nicer than what we could have assembled in a graphics editor, we used this data and plotting software that we had at hand. > Note that this picture is confirmed by e.g. Cokelet et al. , 2008, and several other papers going back a few decades, but we believe you will agree that the situation we depict in Fig. 1 is at least qualitatively fully supported by available literature.

RC: P3, L5-7 You should really provide more evidence that this is now a "universal paradigm" and that the paper you mention above are actually directly and unconditionally transfer results obtained for the Pacific sector to the Atlantic Sector.

AR: > No, it is not (yet). That is why we state "it might appear as if [it is] being cemented", rather than that it "has been" cemented. We also never used the words "directly and unconditionally" (the former only in a different context).

> As for providing evidence that the referenced papers do transfer results from the Pacific side to the Atlantic one, the reader is free to check for themselves; in such a short exposition it would only destroy the flow of the text to quote and paraphrase from all those articles. But see e.g. Våge et al.; "A comparison to hydrographic data from the

Pacific Water boundary current in the Canada Basin under similar atmospheric forcing suggests that upwelling was taking place during the survey." [which took place north of Svalbard, our comment]

> On another note, calling this "a paradigm" does not require that it is occurring everywhere, all the time. Instead, a paradigm is a way of thinking about things. That means that the habit of phrasing shelf break productivity primarily in terms of shelf break upwelling (e.g., Falk-Petersen et al. 2014, Williams&Carmack 2015) we believe qualifies as a paradigm; see also our reply to reviewer 1 on this point. MC: > We amended the sentence to "... currently being cemented as a universal paradigm to conceptualize ..." to make this distinction clearer.

RC: P3, L21 Gradients of what?

AR: > In this context, what matters are the gradients of physical properties, even though there are plenty more. MC: > We added "[large gradients] in physical properties".

RC: Fig. 2 Why you use this transect? Is it typical? Why not climatology or reanalysis?

AR: > Yes, it is typical as it says in the figure caption. Otherwise, it is an illustration, and the same comments apply as for Fig. 1 earlier.

> Just as an example, quoting Falk-Petersen et al. (2014), who were measuring in this area and at the same time of the year, "all transects had very similar hydrographic characteristics, with an upwelling zone of warm Atlantic Water (temperature 3–4 °C, salinity ~35 psu) stretching from west to east along the northern Svalbard Shelf".

> As another illustration, see the attached plot below (showing a transect at a similar location across the shelf break) produced (for wind conditions that people might refer to as neutral in terms of "upwelling-favourability") from the same ROMS 800x800 m model that was used for the right panel of Figure 1. It was made for another (quantitative) manuscript on circumpolar upwelling that is going to be submitted in the not too distant future. We do however choose not to include it in the present manuscript we

are currently discussing in order to not clutter the paper; also based on the available literature (Våge et al. 2016, Cokelet et al. 2008, and others) there should be no doubt that the situation we present in Fig. 2 is representative.

RC: P4, L4-9 It is not clear to me why any reader familiar with Arctic Ocean hydrography must think that Fig. 2 show typical upwelling situation?

AR: > Well, that is exactly one of our points, and we are just as puzzled as you are. Then again, not everyone interested in the biological implications might be sufficiently "familiar" with Arctic Ocean hydrography. Given experiences in other regions of the world ocean, it is at any rate not an abstruse idea to think of upwelling when one sees isolines outcropping at the surface close to a shelf break or coastline.

RC: P4, L13 Ivanov et al., 2016 show that under certain conditions heat from the Atlantic Water can mix up to the surface, but this process is not constant and over the northern Barents Sea shelf the thermal stratification in the upper 100 meters is actually still quite strong most of the time.

AR: > You are right. MC: > We added "potentially [leading to....]" before wintertime convection.

RC: P5 L3-15 Statements in this section need supporting evidence. Now it is pure speculations.

AR: > Assuming by "supporting evidence" you mean references: MC: > We have now inserted a number of references in that section.

RC: P5, L 17 No, we haven't. You just claim it to be true earlier, but you did not show anything to support this claim.

AR: > Specifically, on p.4 l.12-14 (original version) we give two references for how vertical mixing is strong in winter (namely, weak thermal stratification), which directly implies that the mixed layer replenishment of nutrients can happen "without recurrence to wintertime upwelling". That doesn't mean wintertime upwelling is not happening or

cannot ever happen, it just means that "upwelling" is not strictly necessary to replenish nutrients. MC: > We amended the text to "can be replenished" to better express this uncertainty.

RC: I am sorry but most of the rest of the analysis is again just pure speculations and to my opinion have no value as a review.

AR: > See our comments at the outset about intended article type and audience. More details as to what exactly you think is speculation would have been helpful, as certainly not "most" of our statements are. > At any rate, we added many references that you will hopefully find helpful. Also, we did add a sentence to the paragraph explaining why a shelf has to be sufficiently shallow and/or narrow to allow for wind-driven upwelling in order to stress our reasoning there as one of the more central parts of the whole line of arguments.

[Figure]

**Fig. 1.**

---

## Author Comment (AC3) · 19 Dec 2017

[Below, please find: Your comments preceded by RC, "Reviewer Comment", and our replies preceded by AR, "Author Reply" and MC, "Manuscript Change"]

RC: This short commentary manuscript brings up a hot and important topic on how wind- driven upwelling, in conjunction with dramatic sea ice loss, may affect ocean productivity on the vast Arctic continental shelves. The authors argued that "shelf break up- welling is likely not a universal but rather a regional, albeit recurring feature of the new Arctic." I do agree with the authors that regional geographic, atmospheric, oceano-graphic, and sea ice conditions must be taken into account when assessing pan-arctic

upwelling phenomena and their impacts on upper ocean nutrient supply and primary production processes. Nonetheless, the authors can better justify their arguments and greatly improve the paper by providing more concrete data analysis and evidence, particularly in the northern Barents Sea shelf where the authors claim upwelling may function differently from other Arctic shelf systems.

AR: > We appreciate the reviewer's support for our key statement that a number of basic conditions, including geographical, together determine if and how upwelling and associated effects on the ecosystem are important in different parts of the Arctic. As you will see from our replies to the two first reviewers' comments we acknowledge that the purpose of the paper, and therefore also the reason for relying on general examples rather than comprehensive data analysis, should have been more clearly explained in the submitted manuscript. Namely, the goal of this paper is (partly) to show that just by general physical considerations, many conclusions can be reached already. These conclusions can help guide hypotheses and field measurements. In the revised version we have attempted to make the key points clearer; that these basic conditions can and should be assessed for different regions, and that snapshot data don't necessarily allow for underlying mechanisms to be deduced. We hope that our responses and suggested manuscript amendments detailed below are sufficient to clarify these key aspects.

> [Below, please find: Your comments preceded by RC, our replies preceded by AR, and our edits to the text preceded by MC.]

RC: Other comments: Figure 1: The illustrations are too vague and lack important geographic and hydro- graphic features. In the left panel, I would suggest the following changes: (1) add lati- tudinal circles and longitudinal lines, (2) use a better color map to illustrate bathymetry (or at least supplement a color bar for the grayscale), (3) draw general surface and bottom circulation patterns.

MC: > We have now added the broad patterns of surface and Atlantic layer circulation patterns as far as they are relevant to our manuscript; the bottom circulation, however, is not relevant to the manuscript and we did not illustrate it. > The following was added to the figure caption: "Arrows show selected patterns of the general circulation \citep[after][]{polyakov2012warming}. Blue arrows: Pacific-derived and other freshwater flowing along the shelf break, through the Transpolar Drift and in the Beaufort Gyre. Red arrows: Atlantic-derived water entering the Arctic Ocean through Fram Strait and the Barents Sea, flowing along the shelf break, submerging north of the Barents Sea and recirculating along the shelf break through the Arctic Ocean. Other major currents are not indicated here as they are of minor importance to this paper." > We have also added a colorbar and the location of the transect shown later in the manuscript. AR: > We think longitude/latitude coordinates do not contribute significantly in this context, they would rather clutter the figure, which is why we refrain from adding them. If the reviewer has particular reasons for why they should be included we will be happy to reconsider.

RC: In section "Many interconnected phenomena": The authors tried to explain different physical mechanisms that drive upwelling and other physical processes that interact with upwelling. I am uncomfortable that the authors did not attempt to put their discussion in the context of rich literature in upwelling. There is not a single citation in the section, which is unusual.

AR: > You are right, and this was also mentioned by reviewer #2. We have added a fair amount of citations throughout the text to be sure to make it clear to the reader that what we base our argument on is well-established in the literature and not our own speculation.

RC: In section "Drivers of marine productivity vary across the Arctic Ocean": The authors should provide observational evidence when claiming Beaufort Gyre region is "one of most nutrient-depleted regions of the world ocean".

AR: > Agreed; see also our reply immediately above. MC: > We have now inserted Codispoti et al., 2013, as a reference.

RC: Figure 2: Please mark corresponding transect in Fig 1 left panel. Why not plotting temperature, salinity and density fields in this transect all together so that readers can better interpret Atlantic and Arctic water masses, vertical mixing, thermal or haline stratification? How many CTD profiles were casted along this transect? Please mark the CTD cast locations. What were the wind conditions during this transect sampling? I think wind diagnosis would be critical in answering whether or not vertical mixing was caused by upwelling.

AR: > Fig. 2 (Fig 2, right panel in revised version) is an illustration of what the density field looks like, generically, without consideration of special wind situations. Discussing the specifics of this transect would only distract from the general points we are trying to make: That a) these kinds of cross-slope hydrographical snapshot transects do not tell us anything about whether upwelling was happening or not (and so whether we plotted temperature and salinity should not change the reader's judgement anyway), and b) that there is no physical reason to expect a dominant signal. Fig. 2, left panel (previously Fig 1 right panel), illustrates the geographical salinity and temperature patterns, thus indicates water masses present at the surface. MC: > We added a sentence to the figure caption to make clear that this is "just" a representative illustration. > The revised figure also includes station markers now and for completeness' sake bottom bathymetry from IBCAO3 plotted into the transect.

RC: Figure 3: This is not an effective way to illustrate wind patterns. Did black dots repre- sent speeds of east wind component? So only those with 3 m/s or more were showed in the figure? How often were the wind measured? The author stated "only 2% of all summer days through the last 30 years can be considered upwelling-favorable", but what's the sample size in total? It didn't look to me that 30-year wind measurements were included in the analysis given this few data points. To demonstrate seasonal differences in wind patterns and highlight the summer season, a plot of four wind roses that aggregate seasonal wind measurements might be more informative and illustrative.

AR: > Exactly, so the take-home message is that even taking 30 years of wind data only results in so few data points where wind could potentially be upwelling-favourable (*if* the shelf break was shallow enough, see the preceding arguments in our manuscript). A wind rose would not discriminate between short episodes of easterly wind (not sufficiently long to affect Ekman transport) and would hence tend to "overestimate" the occurrence of upwelling-favorable winds.

MC: > We added "[rather low a wind speed and makes for a generous criterion in this regard]; there is no universally accepted measure" to make it clearer to the readers that this methodology is more of a tentative thought experiment, assuming that the wind could drive upwelling in the first place even though the shelf is rather deep in the area in question. The figure caption now also says "assuming the local bathymetry facilitates such upwelling".

RC: In section "Summertime upwelling north of Svalbard?", the argument is unconvincing without showing results from mooring or ship-based hydrographic measurements. Per- sonal communication is not sufficient.

AR: > The argument rests entirely on general physical arguments. The personal communication is just an illustration. MC: > To get our point better across in the manuscript, we inserted "As we have seen, consideration of general physical and geographical patterns alone such as boundary layer physics and wind patterns already leads us to conclude that upwelling should not be expected to feature very prominently on the Barents side of the Arctic. This is not to say that upwelling events cannot ever happen (and indeed, in a system as complex as the Earth, it would be surprising if they would never happen), but no known physical mechanism would suggest a magnitude, frequency or importance similar to what has been found in the Pacific sector. To illustrate our point, let us just mention some upcoming work by A. Renner and collaborators [...]"

RC: In section "Climate Change and the Future of Arctic Marine Productivity", I think another relevant point is the changing phytoplankton abundance and species compo-

sition in response to changing hydrography and nutrients. I would suggest the authors to briefly touch on this point. Two examples are: 1) Li, W. K. W., F. A. McLaughlin, C. Lovejoy, and E. C. Carmack (2009), Smallest algae thrive as the Arctic Ocean freshens, Science, 326, 539; 2) Li, W. K. W., E. C. Carmack, F. A. McLaughlin, R. J. Nelson, and W. J. Williams (2013), Space-for-time substitution in predicting the state of picoplankton and nanoplankton in a changing Arctic Ocean, J. Geophys. Res. Ocean., 118(10), 5750–5759.

AR: > We did not originally include this story about plankton size spectra because our focus has been on the shelf breaks specifically, but we agree that this is important for the large-scale picture. MC: > We have now included them as you suggest; please see last paragraph in the "Climate Change and the Future. . ." section..

---

## Author Response (AR2)

Topic Editor Decision: Reconsider after major revisions (18 Feb 2018) by Mario Hoppema Comments to the Author: Dear Drs. Randelhoff and Sundfjord,

Thanks for your resubmission. I am sorry to convey that I am not satisfied with some of your modifications. I think you did not apply the useful suggestions from the referees to the best extent possible. Please see my comments to that and some further minor comment below. I still think the manuscript is worth publishing and it could convey an important message.

Once again, thank you very much for your efforts in evaluating this manuscript. We are glad you see the value in our manuscript, despite our initial divergence in opinion about the best way to convey some of the elements in it. Below find a detailed response.

Among the major changes, we modified a number of figures according to your suggestions.

In particular, we now added a figure that shows the depth of the Arctic shelf break. This clearly illustrates one of the points we had made in the text: That in most areas of the Arctic Ocean, the shelf break is quite deep and thus a dynamic coupling to wind-driven Ekman transport (hence upwelling) is unlikely. (Again, note how the Chukchi and Laptev seas stand out with rather shallow shelf breaks as opposed to the others.)

Also, we now show a schematic of the wind-driven shelf break upwelling mechanism in another figure (now Figure 3), giving an intuitive picture of what we previously described only in the text.

Furthermore, we now use Mercator data for the surface salinity map in the Svalbard area (formerly Figure 2), as has been initially suggested by one of the reviewers - unfortunately, climatology data was too scarce for our intended use.

Please see more details on these two key points in the point by point responses below, along with the other amendments we have made.

From referee #2 and your response (and also referee #3 touches upon this issue, see below)

> RC: Fig. 1. Why you illustrate Atlantic Water inflow by snapshot from the model, that can be pretty far from reality (Hattermann et al., 2016 do not use data assimilation)? Why not from climatology or some reanalysis product (e.g MERCATOR OCEAN)?

AR: >As you say yourself, it is an illustration, and it does in fact capture the real-world features and patterns that are relevant for this paper (inflow of near-surface warm and salty water). We could have hand-drawn something (this, too, "pretty far from reality", but which would still capture the same essence), but for ease of producing the figure and because it looks nicer than what we could have assembled in a graphics editor, we used this data and plotting software that we had at hand. >Note that this picture is confirmed by e.g. Cokelet et al. , 2008, and several other papers going back a few decades, but we believe you will agree that the situation we depict in Fig. 1 is at least qualitatively fully supported by available literature.

EDITOR: I am not satisfied with your response. First, you only show this at one location in the Barents Sea. Your response is that this is only an illustration. I think an illustration is not sufficient. If you want to show that this situation holds for a larger region (and I think you would like to make this point), you have to show that with data. And secondly, you do not show it based on real data. The referee suggests climatology or a reanalysis product; you did not do anything with this suggestion. I think this is indeed the right data to use here. Actually, my comments especially hold for Figure 2. For showing this it doesn't suffice to use modelled data, only real data take away any doubt.

As you and the reviewer suggest, we now use Mercator ocean reanalysis data for this plot. The main features of the figure as well as the conclusions remain the same, but we agree that using a reanalysis, especially one that assimilates sea ice data, could lend better support to our claims than the ROMS we previously used. (We still think that the ROMS is generally in good agreement with data in this region, but you are right that many readers will be more convinced by a reanalysis.) Text describing the Mercator simulations has now been included in the figure caption.

(We also tried using the climatology MIMOC, but as expected data is too scarce for this area in winter.)

From referee #3

RC: Other comments: Figure 1: The illustrations are too vague and lack important geographic and hydrographic features. In the left panel, I would suggest the following changes: (1) add latitudinal circles and longitudinal lines, (2) use a better color map to illustrate bathymetry (or at least supplement a color bar for the grayscale), (3) draw general surface and bottom circulation patterns.

MC: >We have now added the broad patterns of surface and Atlantic layer circulation patterns as far as they are relevant to our manuscript; the bottom circulation, however, is not relevant to the manuscript and we did not illustrate it. >The following was added to the figure caption: "Arrows show selected patterns of the general circulation citep[after][]{polyakov2012warming}. Blue arrows: Pacific-derived and other freshwater flowing along the shelf break, through the Transpolar Drift and in the Beaufort Gyre. Red arrows: Atlantic-derived water entering the Arctic Ocean through Fram Strait and the Barents Sea, flowing along the shelf break, submerging north of the Barents Sea and recirculating along the shelf break through the Arctic Ocean. Other major currents are not indicated here as they are of minor importance to this paper." >We have also added a colorbar and the location of the transect shown later in the manuscript. AR: >We think longitude/latitude coordinates do not contribute significantly in this context, they would rather clutter the figure, which is why we refrain from adding them. If the reviewer has particular reasons for why they should be included we will be happy to reconsider.

EDITOR: This is again about Figure 1. As referee #1, this referee is not satisfied with Fig 1, and I agree with that. The quality of this Figure is just not high enough. Actually, it is very much distracting from the message. The commentary discusses features along the shelf break, then those shelf breaks should be clearly visible with all of their details. The Figure just does not make a good impression. Please why don't you just provide a high-quality figure with latitude-longitude (of course, every map needs lat-long, no discussion at all). You do not provide a deep data analysis, which is ok as this commentary is not meant to do that, but then at least you can give care to the production of insightful figures, which also enhance the message.

We have now developed a quite detailed map of the Arctic shelf break all along the boundary current, excluding only the Saint Anna Trough and the Chukchi Borderlands. A detailed description and the Python code are included

with the supplementary material.

This map (the new Figure 2) shows clearly that in most areas of the Arctic Ocean, the shelf break itself is actually quite deep most places, except off Alaska and in the Laptev Sea.

Latitudes and longitudes are now included in Fig 1 and the new Figure 2.
* * *
Comment by referee #3 and your response:

> RC: Figure 2: Please mark corresponding transect in Fig 1 left panel. Why not plotting temperature, salinity and density fields in this transect all together so that readers can better interpret Atlantic and Arctic water masses, vertical mixing, thermal or haline stratification? How many CTD profiles were casted along this transect? Please mark the CTD cast locations. What were the wind conditions during this transect sampling? I think wind diagnosis would be critical in answering whether or not vertical mixing was caused by upwelling.

AR: >Fig. 2 (Fig 2, right panel in revised version) is an illustration of what the density field looks like, generically, without consideration of special wind situations. Discussing the specifics of this transect would only distract from the general points we are trying to make: That a) these kinds of cross-slope hydrographical snapshot transects do not tell us anything about whether upwelling was happening or not (and so whether we plotted temperature and salinity should not change the reader's judgement anyway), and b) that there is no physical reason to expect a dominant signal. Fig. 2, left panel (previously Fig 1 right panel), illustrates the geographical salinity and temperature patterns, thus indicates water masses present at the surface. MC: >We added a sentence to the figure caption to make clear that this is "just" a representative illustration. >The revised figure also includes station markers now and for completeness' sake bottom bathymetry from IBCAO3 plotted into the transect.
* * *
EDITOR: I agree with the comments by the referee and think the authors wipe away the valid arguments of the referee too easily. For the reader it is certainly quite useful to see temperature, salinity and density. The authors argue that it should not change the reader's judgement anyway, but please leave that to the reader. For the authors it suffices to show everything that might be important and relevant, explain it and leave the judgment to the reader indeed. Do not care about distraction from the general points you are trying to make: It is especially distracting if obvious factors are left out. So, what about the wind patterns and their analysis? Seems to be a factor when talking about upwelling. At a later stage in the manuscript,

the authors do indeed mention the wind patterns as a factor for upwelling, so why not here?

We have now re-made the transect plot with salinity and temperature following your advice. We also added the wind conditions in the figure text.

The location of the transect is now included in the left-hand panel of that figure, instead of in Fig 1.

From referee #3 and response:

RC: In section "Summertime upwelling north of Svalbard?", the argument is unconvincing without showing results from mooring or ship-based hydrographic measurements. Personal communication is not sufficient.

AR: >The argument rests entirely on general physical arguments. The personal communication is just an illustration. MC: >To get our point better across in the manuscript, we inserted "As we have seen, consideration of general physical and geographical patterns alone such as boundary layer physics and wind patterns already leads us to conclude that upwelling should not be expected to feature very prominently on the Barents side of the Arctic. This is not to say that upwelling events cannot ever happen (and indeed, in a system as complex as the Earth, it would be surprising if they would never happen), but no known physical mechanism would suggest a magnitude, frequency or importance similar to what has been found in the Pacific sector. To illustrate our point, let us just mention some upcoming work by A. Renner and collaborators [...]"

EDITOR: It would be more convincing if the mentioned general physical and geographical patterns were explained so that the reader would understand. Actually, this is exactly what such a review-like paper should convey.

The sentence in question was restructured to "As has been shown above, wind statistics as well as general physical considerations and geographical features - the northern Barents Sea shelf being too deep for surface and bottom Ekman layers to overlap and produce shelf break upwelling - imply that upwelling should not be expected to feature very prominently on the Barents side of the Arctic." in order to make the sentence clearer. Also, we now show a schematic of the wind-driven shelf break upwelling mechanism in yet another figure (now Figure 3), giving an intuitive picture of what we previously described only in the text.

> The abstract does not reflect the contents of the manuscript. It is all about the physical environment and upwelling, but there is hardly any word about what this means to primary productivity. Laying the connection between those two is after all the main goal of the paper. Please restructure the abstract.

We added the following to close the loop back to the beginning of the abstract and make the connection to the biology, as you suggest: " Still, other factors can contribute to marked future increases in biological productivity along the Arctic shelf break. A warming inflow of nutrient-rich Atlantic Water feeds plankton at the same time as it melts the sea ice, permitting increased photosynthesis. Concurrent changes in sea ice cover and zooplankton communities advected with the boundary currents make for a complex mosaic of regulating factors that do not allow for Arctic-wide generalizations."

> P1, L18 Many regions, but only one reference. Please give some more.

The reference is to the entire recent book by Kämpf and Chapman on upwelling systems, which has chapters on almost every region of the global ocean. A literature search on our side did not yield review articles with a similar scope, and any regional study would pale in comparison.

> P1, L22 What is ibid here? It is definitely clearer to give the correct citation here.

The reference is now made explicit as the textbook by Kämpf and Chapman.

> P2, L1 I think this is the right place to give at least several references, and possibly add: "and more below")

We followed your suggestion.

> P2, L2 "As the ice cover recedes from the shelves into the basin" It is not clear what time scale is valid here, seasonal, annual, decadal? Maybe rephrase to make this clear.

The text now specifies that we mean interannual time scales.

> P2, L3 "primary production is projected to keep increasing (Arrigo and van Dijken, 2015):" Some reasons for higher production are mentioned. There are also factors that may cause less production, e.g. enhanced stratification. Since it is not clear at all whether the production would increase, I think the entire story should be told here. Moreover, only one study with a positive response is cited here. Since the increase of primary production is a central point in your argument, you need to refer to more studies that may predict higher production. If at all known, also studies that project no increase in

production should be cited.

We rephrased this as "has been observed" to take away the focus from the projection into the future which is, as you indicate, a much more controversial issue than the satellite-based overall increases in new primary production.

We have also clarified in the text "net" primary production to make this distinction clear, and inserted two more references. (The "net" PP distinction is important because new production is something else and might possibly go the other way as stratification increases, as you remark.)

> P2, L9 "and argued that decreased ice concentrations will enhance upwelling in the area" For what reason did they argue that? That should be important info for the reader here.

The text now clarifies their reasoning.

> Figure 1 caption L2: delete: black (word is not necessary, and giving the wrong associations)

We did as you suggested, and replaced by "the box in dashed lines"

> P4, L2 delete: Not surprisingly (not appropriate here)

We followed your suggestion.

> P4, L3 in the Arctic Ocean (instead of in the world ocean)?

I think it is safe to say world ocean here; we added two more references to show that.

> Figure 2 In such a figure, latitude and longitude is of much help to the reader and it is standard indeed. For example, it would be visible where the section is situated. The cruise data presented in the second panel need a reference.

[tbd] The updated figure now includes the coordinates. We have added reference to the data set, which has been published and has a doi.

> I agree with referees # 2 and # 3 that it would be more correct to take real data instead of modelled data. Is there any evidence that the larger-scale situation is exactly in agreement with this model? In a situation like this, real data, if available, is always preferred above modelled data.

As discussed above, we now use the Mercator Ocean reanalysis for this plot.

P5, L4-5 "In contrast, the Beaufort sea is strongly stratified throughout the year; there, winter upwelling can be an important factor contributing to the pre-bloom nutrient pool" Earlier it was stated (and the data from Codispoti et al show it as well) that the nutrient concentration, even in winter are very low, i.e. not much productivity there, and so upwelling does not enhance productivity in reality.

We now see that our original sentence was ambiguous - the revised version is explicit that we mean that potentially increasing winter upwelling (because of the receding ice edge etc.) would enhance the pre-bloom nutrient reservoir.

P5, L22 "This is because water that already is at the surface will not profit from further upwelling" This is not unequivocal. The characteristics of this water at the surface should at least be added.

Our original sentence was awkward and prone to be misunderstood; we deleted it in the revised version because it did not contribute much to the paragraph.

P5, L28 delete: as we will see later

We did as you suggested.

Figure 3: There is also a comments by referee #3 about this. What surprises me is the piece of text: "based on ERA-INTERIM data". Does this mean that those data, and all of those data, were used? But then, the reference is from 2011, while the data should be for 1987-2017?

ERA-Interim is operational and updated contiuously; the reference we give is the one describing their assimilation system, officially endorsed on their web site (`https://www.ecmwf.int/en/forecasts/datasets/reanalysis-datasets/era-inte`

In the caption of Figure 3 "A daily windspeed was considered "potentially upwelling-favourable" if its (approximately easterly) along-shelf component exceeded 3 m s-1 for at least 3 consecutive days. (3 m s-1 is rather low a wind speed and makes for a generous criterion in this regard; there is no universally accepted measure.)" First, I wonder where the criterion comes from. There must be some literature about such issues, I think. And then second, if the criterion is not generally accepted, as you write, this strongly reduces the validity of your argument. As to the few appropriate data, could it be that the model data and reanalysis data are biased with regard to along-shelf winds?

There is of course literature about this, nicely summarized in Chapter 2.1 of the book by Kämpf and Chapman (albeit for the case of coastal upwelling). The upshot is that wind speeds of 5 m/s and durations of 5 days would be appropriate numbers in a typical coastal upwelling area at lower latitudes, but we devided to err on the "upwelling-favourable" side to make our point clearer (that the appropriate winds are rare) and not leave the readers wondering whether relaxing the criterion would have made a huge difference.

The figure text of Figure number 4 (formerly 3) is now changed accordingly.

As for the potential bias you mention, we are not aware of any in this regard, and the ERA-Interim web site (`https://software.ecmwf.int/wiki/display/CKB/ERA-Interim+known+issues`) does not state any such issues either. However, if you can point us to indications of such a bias, we are happy to take a closer look and discuss it in the manuscript as needed.

> P7, L1-2 "But it should be kept in mind that ice cover by itself is not a show stopper for wind driven upwelling (or for Ekman pumping for that sake)" The connection to the previous sentence could be guessed at most. I think the steps you are making are too big. Please rephrase clearly what you intend to convey here.

We reworded this sentence: "But it should be kept in mind that the mere earlier presence of an ice cover would not have prohibited wind driven upwelling (or Ekman pumping for that sake), and could even have enhanced it in some circumstances."

In addition to the changes described above, we have made some minor adjustments to wording in a few places. None of these change the substance, but should ease reading or add precision. All changes are visible in the tracked changes version.

[revised manuscript text omitted]